# A Comparative Study of Metabolic Syndrome Using NCEP—ATP III and IDF Criteria in Children and Its Relationship with Biochemical Indicators in Huatusco, Veracruz, Mexico

**DOI:** 10.3390/children10030473

**Published:** 2023-02-27

**Authors:** Eduardo Rivadeneyra-Domínguez, Joel Jahaziel Díaz-Vallejo, Aurora Guadalupe Prado-Bobadilla, Juan Francisco Rodríguez-Landa

**Affiliations:** 1Facultad de Química Farmacéutica Biológica, Universidad Veracruzana, Xalapa 91000, Veracruz, Mexico; 2Laboratorio de Neurofarmacología, Instituto de Neuroetología, Universidad Veracruzana, Xalapa 91190, Veracruz, Mexico

**Keywords:** metabolic syndrome, insulin resistance, obesity, overweight

## Abstract

Metabolic syndrome includes a set of metabolic alterations associated with overweight and obesity. The criteria for its diagnosis are heterogeneous, and there have been few studies about prevalence in the pediatric population. The aim of this study was to describe how the estimated prevalence of metabolic syndrome varies by International Diabetes Federation (IDF) vs. National Cholesterol Education Program—Adult Treatment Panel III (NCEP—ATP) criteria. We conducted a cross-sectional study in which anthropometric information, triglyceride, cholesterol, glycemia, and insulin levels, among others, were collected. We compared the group potentially misclassified by IDF with the group classified without metabolic syndrome by NCEP—ATP with respect to weight status and biomarkers. Statistical analysis included linear regression, Mann–Whitney U test, Fisher´s exact test, and odds ratio calculation. The IDF criteria missed the association with obesity, although the undetected group differed significantly from the nonmetabolic syndrome group in terms of IBM and weight. The associated biomarkers were ultrasensitive C-reactive protein (Hs-CRP), alanine aminotransferase (ALT) enzyme, insulin, triglycerides, and high-density lipoprotein (HDL) cholesterol. Waist circumference was the parameter with the strongest association for presenting metabolic syndrome, with an odds ratio of 18.33. The results of this study showed the estimated prevalence of MS varies by criteria, due to cutoff points, and how the high prevalence of MS strongly associated with obesity.

## 1. Introduction

Metabolic syndrome (MS) involves a set of metabolic disturbances including insulin resistance, abdominal obesity, increased fasting glucose, hypertension, hypertriglyceridemia, hypercholesterolemia, proinflammatory state, and low high-density lipoprotein (HDL) concentration, among others [1]. The global prevalence of MS in the adult population is estimated to be 20–40% [2]. It is one of the major risk factors for the development of type 2 diabetes mellitus and cardiovascular disease, and it is associated with the global epidemic of overweight and obesity [3].

As in adults, the prevalence of overweight and obesity in children and adolescents has been increasing. According to the World Health Organization (WHO), it increased from 8% in 1975 to 18% in 2016 [4,5]. In Mexico, according to the results of the 2021 National Health and Nutrition Survey, the combined prevalence of overweight and obesity was 37.4% in children aged 5 to 11 years and 42.9% in adolescents aged 12 to 19 years [6], making it one of the main public health problems. The pathogenesis of MS is complex, but insulin resistance and obesity increase the risk of developing it, so the presence of obesity and overweight in children and adolescents is a risk factor for this condition. Furthermore, overweight in childhood and adolescence is associated with an increased risk and earlier onset of type 2 diabetes [7]. 

The diagnosis of MS has generated controversy due to a lack of consensus on its definition for children and adolescents worldwide, which is why the International Diabetes Federation (IDF) adapted a classification for this population group [8]. Another widely accepted classification is that proposed by the National Cholesterol Education Program—Adult Treatment Panel III (NCEP—ATP) [9,10], see Table 1. However, there are no standardized criteria worldwide. This has led several authors to modify the criteria and cutoff points for the diagnosis of MS. In Mexico, there have been few studies of MS in children and adolescents, which is why information is limited and it has been difficult to establish criteria between age groups and BMI for diagnosing obesity and MS. The aim of this study was to describe how the estimated prevalence of MS varies by IDF vs. NCEP—ATP criteria and how the relationship between MS and obesity and chronic disease risk factors varies by IDF vs. NCEP—ATP criteria in a sample of school-aged children and adolescents from a community in Veracruz, Mexico.

## 2. Materials and Methods

### 2.1. Study Design and Participants

A cross-sectional study was conducted in Mexican children and adolescents aged 8–15 years with a random sample of students from two public educational institutions in the town of Huatusco, Veracruz, Mexico. The schools are urban and low-income, and they were selected for convenience of access and location. The school enrollment of third to sixth graders in each school was 92 and 87 children, respectively.

For the selection of participants, a sample size calculation was performed with an expected proportion of 10% SM according to previous studies [11]. Individuals were randomly selected according to the list of students from both schools, with the same probability of selection, receiving a direct invitation to participate and the subsequent voluntary acceptance of the child´s guardian. From the sample, a total of 91 children were eligible, two children were excluded because of a previous medical diagnosis of diabetes mellitus and heart disease; 17 children refused to participate who did not differ from the characteristics of the children included in the study. In total, 72 participants were included. Interviews and measurements were conducted in the months of September 2020 to March 2021.

### 2.2. Measures

#### 2.2.1. Anthropometric Characteristics

A brief survey was conducted on sociodemographic characteristics and family history for data collection. The survey was designed by a pediatric physician and a nutritionist. Anthropometric measurements were taken in the presence of the parents and/or child´s guardian, and American Union criteria were used. Weight was measured using a scale and SECA^®^ stadiometer while barefoot and wearing light clothing, and abdominal circumference was measured with an inextensible tape measure. Waist circumference was measured at the lower edge of the last rib and the upper edge of the iliac crest; the standard error for weight was less than 0.02 kg, and that for circumference was less than 0.01 cm, as indicated by the American Diabetes Association. Body mass index (BMI) was calculated for nutritional status using the Z-score with its respective cutoff points according to the WHO [12]. Trained personnel measured blood pressure with a sphygmomanometer (Welch Allyn, Ciudad de Mexico, Mexico) and stethoscope (3M^TM^Littmann, Ciudad de Mexico, Mexico).

#### 2.2.2. Clinical Samples and Biomarker Measurement

Sample extraction and analysis were carried out at the Huatusco Clinical Specialties Laboratory (Av. 2 no. 616 between street 4 and street 6, CP. 94100 Huatusco de Chicuellar, Veracruz, México), the same place where, owing to the COVID-19 contingency, anthropometric measurements and phlebotomies were performed under sanitary conditions in the presence of parents and/or child´s guardian. Blood samples were obtained via venous puncture (cephalic, mid-ulnar, and basilic vein) with a prior fasting period of 10–12 h; the volume collected per child was approximately 8–10 mL.

For the determination of blood chemistry and liver function, a Wiener Lab CM250 automatic analyzer (version 4.2 MT-Promedt Consulting GmbHaltenhofstr. 80D-66386 St. Ingbert/Germany, colorimetric method) was used. For alanine aminotransferase (ALT), aspartate aminotransferase (AST), and gamma-glutamyl transferase (GGT) determinations, enzymatic methods were used. For the analysis of insulin and ultrasensitive C-reactive protein (hs-CRP), we used the IMMULITE 1000 (Siemens Healthcare Diagnostics Inc., Flanders, NJ. 078369657, USA) automated analyzer.

##### Definitions

Insulin Resistance: insulin resistance was assessed with the HOMA-IR (Homeostatic Assessment Model) method using the following formula: fasting insulin (μU/mL) × basal glucose (mg/dL)/405, with a cutoff point at values equal to or greater than 2.86, according to previous studies in the Mexican population [13]. For Metabolic Syndrome: the criteria adapted for children and adolescents from NCEP—ATP III (National Cholesterol Education Program—Adult Treatment Panel III) [9,10] and IDF (International Diabetes Federation) [8] were used. MS was considered when three or more conditions were present. These conditions are shown in Table 1.

### 2.3. Statistical Analysis

The prevalence of MS was estimated using the NCEP—ATP and IDF criteria. The agreement between classifications was determined. We compared the group potentially misclassified by IDF with the group classified as without MS by NCEP—ATP with respect to weight status and biomarkers.

Tests for the normality of variables were performed, and descriptive statistics were used for quantitative variables according to distribution. Mann–Whitney U test for median comparisons was used. Fisher´s exact test was used to compare categorical variables. We used linear regression to analyze differences between groups. Mean and standard deviation were calculated. Odds ratios (ORs) with 95% confidence intervals (95% CIs) were estimated. The significance level was set as *p* < 0.05. The Statistical Package for Social Sciences (SPSS-IBM) version 25 was used.

## 3. Results

A total of 72 students participated in the study, with a median age of 10 years. They were 54.2% male (*n* = 39) and 45.8% female (*n* = 33). Table 2 shows the characteristics of the study population; differences were found in the anthropometric variables of weight, BMI, and age, according to criteria for MS. 

In the analysis by age group and sex, significant differences were found in the 8–9 years age group between girls and boys for weight (27.16 and 40.60 kg, respectively; *p* = 0.001), waist circumference (59 and 78 cm, respectively; *p* = 0.001), and BMI, which was lower in girls. Regarding biomarkers, there was a significant difference in basal glucose, which was higher in girls (94 mg/dL) than in boys (87.5 mg/dL). For Hs-CRP, the concentration was higher in boys (0.91 mg/dL) than in girls (0.15 mg/dL). In the 10–15 years age group, no significant differences were found. Regarding genetic factors, 48.6% of the children had no family history of diabetes and 51.4% mentioned having a history of type 2 diabetes mellitus in at least one member of their family, which was corroborated by their child´s guardian. On the other hand, 19.12% of the children reported having a family history of hypertension, 79.5% reported having no family history of hypertension, and 1.38% did not know this information. A family history of obesity was reported at a frequency of 83.34%, and no associations were found with respect to MS.

The combined prevalence of overweight and obesity was 54.2%, with a significantly higher rate of obesity in boys than in girls, as shown in Table 3. The prevalence of MS according to ATP III criteria was 22.2% (12.4–32.1%, 95% CI), and no sex differences were found, with 23.1% of males (9.2–36.9%, 95% CI) and 21.2% of females (6.5–35.9%). Using the IDF criteria, the prevalence was found to be 13.9% (5.7–22.1%, 95% CI), with no sex difference. Of the participants, 23.6% had a waist circumference >90th percentile (P90), 26.4% had blood pressure > P90, 47.2% had triglycerides > 110 mg/dL, 51.4% had HDL < 40 mg/dL, 9.7% had basal blood glucose > 110 mg/dL, and 37.5% had insulin resistance according to HOMA. 

Table 4 presents the nutritional status according to BMI and the components of MS corresponding to ATP III and IDF criteria, showing significant differences in the prevalence of MS, which was higher in the obese group (44.4%) compared with the overweight (16.7%) and normal weight (6.1%) groups in the case of ATP III criteria; using the IDF criteria, this difference was not significant. A significant association was found between obesity and MS by the NCEP—ATP III criteria (OR: 8.2, 95%CI: 2.3–29.4), which was not detected using the IDF criteria (OR: 2.9, 95%CI: 0.7–11.5). No statistically significant differences were found when analyzed by sex.

Table 5 shows the significant differences found between the groups with and without metabolic syndrome with respect to biomarkers; uric acid, ALT, insulin, and Hs-CRP levels were higher in the MS group.

Table 6 shows the estimated ORs; with respect to biomarkers, an association was found between Hs-CRP above the normal value and metabolic syndrome, with OR = 3.66 (1.11–12.02, CI 95%) (*p* = 0.027).

## 4. Discussion

The prevalence of MS has varied in different studies carried out in the pediatric population due to the criteria and cutoff points used by the authors, which makes comparisons between studies difficult. In our study, the prevalence varied when using the NCEP—ATP III and IDF criteria, as we obtained a lower prevalence with the latter. 

Previous studies in Mexico have reported a higher prevalence of MS. For example, a study conducted in a primary care medical unit in a population aged 6 to 16 years, using the NCEP—ATP III criteria, estimated a prevalence of 33% [14]. A higher prevalence of 40% was reported in school children aged 6 to 12 years, and a prevalence of 54.6% was reported in children with obesity [15]; in the present study, the prevalence in the obese population was 44.4%. 

Regarding nutritional status, a significant difference was observed in the prevalence of MS, being higher in children with obesity when using the ATP III criteria. With the IDF criteria, despite also estimating a higher prevalence in children with obesity, it was not statistically significant as compared with the other nutritional statuses. The IDF definition failed to identify the remaining 8% that were identified with the ATP III criteria. This is because the IDF criteria increase their cutoff points for blood pressure and triglycerides that normally apply to adults, so that children with high values but not above the cutoff point, are discarded and inappropriately classified, possibly underestimating prevalence.

It is important to mention that in the present study, insulin resistance was found in 37% of the obese group, which was lower than the 65% previously reported in Mexican obese children [16]. The authors of that study also found a high percentage of insulin resistance in children without a diagnosis of MS. In our study, we found that 33.9% of the children, despite not yet meeting the ATP III criteria for MS classification, were in a possible predisposing state for the development of the disease. In addition, obesity is defined as abnormal or excessive fat accumulation that increases the risk of developing a secondary disease. Adipose tissue was previously considered as a static tissue (reservoir for energy). Studies have referred to adipose tissue as a dynamic tissue (metabolically active organ) [17,18,19]. The morphophysiological change of adipose tissue during obesity induces a chronic low-grade inflammatory state, also referred to as parainflammation (intermediate state between basal and inflammatory) or metainflammation (metabolically triggered inflammation) [20,21,22]. Many studies report that, during this inflammatory process, there is excessive segregation of inflammatory factors known as adipokines, bioactive molecules involved in the etiology of inflammation and insulin resistance associated with obesity [23], segregated by adipocytes that include TNF, IL-6, IFN-, plasminogen activator inhibitor (PAI-1), monocyte chemoattractant protein-1 (MCP1), IL-1, IL-8, IL-10, IL-15, leukemia inhibitory factor (LIF), hepatocyte growth factor (HGF), serum apolipoprotein amyloid A3 serum (SAA3), macrophage migration inhibitory factor (MIF), potent inflammatory modulators (such as leptin, adiponectin, resistin), and high sensitive C-reactive protein (hs-CRP), and these maintain both negative and positive effects, such as the maintenance of oxidative stress, changes in autophagy patterns, and tissue necrosis, principally [24,25]. On the other hand, highly sensitive C-reactive protein is a reactant, that is, a plasmatic protein that undergoes alterations in phases of inflammation, which is synthesized by the liver and deposited in sites with inflammatory processes. Recent studies have shown that the intima of arteries with atherogenesis as cardiovascular disease (CVD) even before any clinical manifestation, accompanied by tumor necrosis factor and interleukins [26,27]. hs-CRP, which is used as a biomarker for the diagnosis of MS, was found to be associated with an OR of 3.66 (1.11–12.02, 95% CI). In this regard, several studies have reported the usefulness of this biomarker as an indicator of an inflammatory process related to the development of MS and diabetes mellitus [28,29].

Regarding serum transaminases, only ALT with values above the biological reference ranges showed an association with MS, with an OR of 3.57 (1.07–11.89 CI 95%). Other studies have suggested that elevated transaminase concentrations are associated with obesity and may be early markers of MS [30,31]. Therefore, it is important to incorporate components that have been associated with the risk of developing MS such as insulin resistance, hs-CRP, and transaminases, and to assess the possibility of improving the prognosis of the condition, during treatment.

The mean serum levels of other biomarkers, such as creatinine (0.69 mg/dL), were found to be within the biological reference ranges. In the case of uric acid, levels below the cutoff point were found without any association with MS, unlike in other studies where hyperuricemia has been observed as an alteration in MS [32].

It is important to highlight some limitations of the present study. First, the incorporation of variables related to healthy lifestyles is lacking, which could be used to perform more in-depth analyses of the associations. Second, the small sample size, which was smaller than that reported in other studies; data collection was conducted during the COVID-19 pandemic and some householders did not accept the invitation for their children to participate in the present study. The small sample size may have influenced the fact that some biomarkers were not found to be related to MS, as previously reported. Furthermore, owing to the cross-sectional design of the study, causality could not be inferred. A further limitation was the lack of biomarkers that evidence low-grade systemic inflammation such as tumor necrosis factor alpha (TNF-α) and interleukins (IL-1,6,17), which are frequently used in clinical research, but are not routinely included in the clinical laboratory due to their high cost. 

Although our population and age cutoff may have some limitations, it gives rise to consideration in future studies for including a population with a larger number of individuals and a wider age range according to the WHO classification of infants and adolescents, with the aim of expanding and looking for other associated factors. 

An advantage of the study was to work with data from school children, from which it was possible to estimate the prevalence of MS in different nutritional states, where children could be considered healthy, unlike research conducted in a hospital environment, where there could be a bias and overestimation of prevalence because children who are brought by their child´s guardian may already have some pathology and even in advanced stages of the disease due to the lack of a culture of prevention. 

The results of the present study provide insight into the most relevant components associated with MS and which biomarkers could be used for early diagnosis. We believe that the estimates obtained in this study warrant further work to determine how to infer a child’s risk of developing MS from weight status and take action to prevent metabolic syndrome.

The prevalence of metabolic syndrome varies across studies [11], depending on the criteria used to define MS. Failure to detect children with preclinical risk factors misses intervention opportunity in the window of prevention. To inform choice of MS criteria to reduce chronic disease risks for children in Mexico, this paper explored the difference between MS classification based on IDF and NCEP—ATP III.

## 5. Conclusions

The results of this study showed the estimated prevalence of MS varies by criteria, owing to cutoff points, and how the high prevalence of MS strongly associated with obesity. Children in this study with obesity and overweight have a high percentage risk of developing cardiovascular disease and type 2 diabetes mellitus in adulthood. Therefore, the use of biomarkers in the pediatric population for the timely diagnosis of MS may be a useful tool when adopting strict measures to prevent the development of other conditions. Additional studies need to be conducted to identify other possible factors. It is also necessary to design school-based preventive strategies through early detection, promotion of healthy lifestyles, and physical activity.

MS is a condition caused by multiple factors, most of which are preventable, such as obesity. Therefore, it is necessary to implement comprehensive public policies to prevent obesity in children and adolescents at home, school, and community.

## Figures and Tables

**Table 1 children-10-00473-t001:** Diagnostic criteria for metabolic syndrome in the pediatric population.

Parameter	NCEP—ATP III Criteria	IDF Criteria
Waist circumference	≥percentile 90	≥percentile 90
Blood pressure	≥percentile 90	≥130/85 mmHg
Triglycerides	≥110 mg/dL	≥150 mg/dL
HDL cholesterol	≤40 mg/dL	≤40 mg/dL
Basal glucose	≥100 mg/dL	≥100 mg/dL

NCEP—ATP III = National Cholesterol Education Program—Adult Treatment Panel III. IDF = International Diabetes Federation.

**Table 2 children-10-00473-t002:** Anthropometric characteristics and biomarkers of children and adolescents in Huatusco, Veracruz, Mexico.

Characteristics	Participants Mean ± SD *n*= 72	No MS by both Criteria Mean ± SD *n*= 56	No MS by IDF, but MS by NCEP—ATP III Mean ± SD *n*= 6	MS by both Criteria Mean ± SD *n* = 10	*p*-Value *		
					a	b	c
Anthropometric							
Weight (Z-score)	0.8 ± 1.2	0.9 ± 1.2	2.2 ± 0.7	1.1 ± 1.3	0.003 **	0.255	0.077
Height (Z-score)	0.3 ± 1.1	0.4 ± 1.2	0.7 ± 1.5	0.2± 1.3	0.350	0.988	0.431
BMI (Z-score)	0.8 ± 1.3	0.9 ± 1.3	2.1 ± 0.5	2.5 ± 1.3	0.005 **	0.100	0.194
Waist circumference (cm)	73.6 ± 12.2	71.1 ± 11.0	88.0 ± 5.1	79.1 ± 14.5	0.001 **	0.043 **	0.130
Age (years)	10.7 ± 2.3	10.8 ± 2.2	10.0 ± 1.8	10.8 ± 2.9	0.420	0.996	0.504
Blood pressure							
Systole (mmHg)	108.4 ± 9.6	106.4 ± 8.8	116.7 ± 10.3	114.5 ± 9.6	0.010 **	0.011 **	0.642
Diastole (mmHg)	67.5 ± 7.3	65.9 ± 6.8	72.8 ± 4.7	73.1 ± 7.3	0.020 **	0.003 **	0.939
Metabolic							
Glucose (mg/dL)	89.9 ± 8.2	89.3± 7.6	89.0± 5.3	93.2 ± 11.6	0.923	0.173	0.323
Urea (mg/dL)	22.6 ± 6.4	22.3 ± 6.5	23.8 ± 5.3	23.3 ± 7.1	0.583	0.655	0.873
Creatinine (mg/dL)	0.7 ± 0.2	0.7 ± 0.2	0.7 ± 0.2	0.7 ± 0.2	0.760	0.948	0.833
Uric acid (md/dL)	3.8 ± 1.2	3.6 ± 1.1	4.5 ± 1.6	4.3 ± 1.0	0.084	0.063	0.839
Triglycerides (mg/dL)	119.0 ± 63.9	107.4 ± 52.4	131.7 ± 12.0	176.6 ± 103.4	0.349	0.001 **	0.151
HDL (mg/dL)	39.3 ± 12.1	40.9 ± 11.6	38.2 ± 15.7	30.4 ± 9.4	0.581	0.011 **	0.204
AST (U/L)	25.5 ± 11.6	25.0 ± 9.5	25.5 ± 5.0	28.5 ± 21.7	0.921	0.385	0.620
ALT (U/L)	24.3 ± 24.6	20.1 ± 15.9	28.3 ± 11.4	45.4 ± 50.5	0.414	0.002 **	0.161
GGT (U/L)	18.8 ± 8.5	18.6 ± 7.8	19.2 ± 10.8	19.6 ± 11.5	0.884	0.742	0.923
Hs-CRP (mg/dL)	1.3 ± 1.6	1.1 ± 1.5	2.3 ± 2.4	1.5 ± 1.4	0.081	0.466	0.328
Insulin (µUI/mL)	13.8 ± 13.9	11.3 ± 6.1	18.6 ± 9.7	24.5 ± 32.6	0.209	0.005 **	0.388
HOMA	3.2 ± 3.9	2.5 ± 1.4	4.2 ± 2.4	6.3 ± 9.4	0.305	0.004 **	0.271

* Linear regression to compare the groups: a. No MS by both criteria vs. No MS by IDF but MS by NCEP—ATP III; b. No MS by both criteria vs. MS by both criteria; c. No MS by IDF but MS by NCEP—ATP III vs. MS by both criteria; ** *p* < 0.05; SD = standard deviation; MS = metabolic syndrome; NCEP—ATP III = National Cholesterol Education Program—Adult Treatment Panel III; IDF = International Diabetes Federation; BMI = body mass index; HDL = high-density cholesterol; ALT = alanine aminotransferase; AST = aspartate aminotransferase; GGT = gamma-glutamyl transferase; Hs-CRP = ultrasensitive C-reactive protein; HOMA = Homeostasis Assessment Model.

**Table 3 children-10-00473-t003:** Nutritional status of the study population according to BMI Z-score in Huatusco, Veracruz, Mexico.

	Male (*n* = 39)		Female (*n* = 33)		Total (*n* = 72)		
Nutritional Status	*n*	% (CI 95%)	*n*	% (CI 95%)	*n*	% (CI 95%)	*p*-Value *
Normal weight	11	28.2 (13.4–43.0)	22	66.7 (49.7–83.6)	33	45.8 (34.0–57.6)	0.001 **
Overweight	6	15.4 (3.5–27.2)	6	18.2 (4.3–32.1)	12	16.7 (7.8–25.5)	0.762
Obesity	22	56.4 (40.1–72.7)	5	15.2 (2.2–28.1)	27	37.5 (26.0–49.0)	<0.001 **

* Fisher´s exact test; ** *p* < 0.05.

**Table 4 children-10-00473-t004:** Components of metabolic syndrome according to nutritional status in Huatusco, Veracruz, Mexico.

Parameters	Normal Weight *n* = 33	Overweight *n* = 12	Obesity *n* = 27	*p*-Value *
	*n* (%)	*n* (%)	*n* (%)	
Waist circumference ≥ P90	0	3 (25.0)	14 (51.9)	<0.001 **
Blood pressure ≥ P90	5 (15.2)	2 (16.7)	12 (44.4)	0.034 **
Blood pressure > 130/85 mmHg	0	0	1 (3.7)	0.542
Triglycerides ≥ 110 mg/dL	12 (36.4)	8 (66.7)	14 (51.9)	0.163
Triglycerides > 150 mg/dL	17 (51.5)	6 (50)	14 (51.8)	0.994
HDL cholesterol ≤ 40 mg/dL	17 (51.5)	6 (50.0)	14 (51.9)	1.000
Basal glucose ≥ 110 mg/dL	3 (9.1)	2 (16.7)	2 (7.4)	0.753
HOMA ≥ 2.9	11 (33.3)	6 (50.0)	10 (37.0)	0.586
Metabolic syndrome (ATP III)	2 (6.1)	2 (16.7)	12 (44.4)	0.001 **
Metabolic syndrome (IDF)	2 (6.1)	2 (16.7)	6 (22.2)	0.174

* Fisher´s exact test; ** *p* < 0.05; P90 = 90th percentile; HOMA= Homeostasis Assessment Model.

**Table 5 children-10-00473-t005:** Biomarkers and metabolic syndrome in Huatusco, Veracruz, Mexico.

	MS by NCEP—ATP III		
Biomarkers	Yes Mean ± SD	No Mean ± SD	*p*-Value *
Glucose (mg/dL)	91.6 ± 9.7	89.3 ± 7.7	0.524
Uric acid (md/dL)	4.4 ± 1.3	3.6 ± 1.1	0.032 **
Cholesterol (mg/dL)	136. 9 ± 49.7	142.2 ± 38.1	0.684
HDL (mg/dL)	33.3 ± 12.3	40.9 ± 11.6	0.022 **
Urea (mg/dL)	23.5 ± 6.3	22.3 ± 6.5	0. 704
Creatinine (mg/dL)	0.7 ± 0.2	0.7 ± 0.2	1.000
Triglycerides (mg/dL)	159.8 ± 83.5	107.4 ± 52.4	0.004 **
AST (U/L)	27.4 ± 17.1	25.0 ± 9.6	0.760
ALT (U/L)	39.0 ± 40.6	20.1 ± 15.9	0.025 **
GGT (U/L)	19.4 ± 10.9	18.6 ± 7.8	0.892
Hs-CRP (mg/dL)	1.8 ± 1.8	1.1 ± 1.5	0.048 **
Insulin (µUI/mL)	22.3 ± 26.1	11.3 ± 6.1	0.043 **
HOMA	5.5 ± 7.5	2.5 ± 1.4	0.055

* Mann–Whitney U tests; ** *p* < 0.05; SD = standard deviation; HDL = high-density cholesterol; ALT = alanine aminotransferase; AST = aspartate aminotransferase; GGT = gamma-glutamyl transferase; Hs-CRP: ultrasensitive C-reactive protein; HOMA = Homeostasis Assessment Model.

**Table 6 children-10-00473-t006:** Estimation of odds ratios (OR) for metabolic syndrome by NCEP—ATP III.

Parameters	OR (CI 95%)	*p*-Value
HOMA ≥ 2.9	1.9 (0.6–6.0)	0.244
Waist circumference ≥ P90	18.3 (4.7–71.0)	<0.001 **
Blood pressure ≥ P90	8.7 (2.5–30.0)	<0.001 **
Triglycerides ≥ 110 mg/dL	7.2 (1.8–28.3)	0.002 **
HDL cholesterol ≤ 40 mg/dL	10.0 (2.1–48.5)	0.001 **
Insulin > 17.0 µUI/mL	4.0 (1.2–12.7)	0.019 **
Basal glucose ≥ 110 mg/dL	3 (0.6–15.1)	0.169
Obesity	8.2 (2.3–29.4)	<0.001 **
Creatinine >1.0 mg/dL	1.8 (0.2–21.2)	0.630
Urea mg/dL	1.6 (0.4–7.1)	0.520
AST > 24 U/L	0.6 (0.2–1.7)	0.316
ALT > 23 U/L	3.6 (1.1–11.9)	0.032 **
Uric acid > 5 mg/dL	1.2 (0.2–6.6)	0.840
GGT > 32 mg/dL	3.9 (0.5–28.8)	0.172
Hs-CRP > 1.0 mg/dL	3.7 (1.1–12.0)	0.027 **

** *p* < 0.05, HDL = high-density cholesterol; ALT = alanine aminotransferase; AST = aspartate aminotransferase; GGT = gamma-glutamyl transferase; Hs-CRP = ultrasensitive C-reactive protein.

## Data Availability

Not applicable.

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
