# Peer review of "A Comparative Study of Metabolic Syndrome Using NCEP—ATP III and IDF Criteria in Children and Its Relationship with Biochemical Indicators in Huatusco, Veracruz, Mexico"

_children, 2023, doi:10.3390/children10030473_

Round 1

Reviewer 1 Report

In this paper the authors report on the prevalence of the metabolic syndrome in children in a small sample on two schools in Huatusco, Mexico.

To my opinion the paper is more of interest for the local (Mexican) readership than the international readership, because of the small sample size and the possible selection bias.

The title does not represent the content. Children are not mentioned in the title and it is not the state that has been studied, but just two schools in one city.

My greatest concerns is the rather small sample size and the fact that in the methods the way the study inclusion was organised have not been described.

It is not clear have the children were recruited and how many children were eligible. It could surely be the case that only children with a problem were more prone to join the study. Furthermore these two schools surely won't be representative for the state, which will include rural areas.

A lot of data are presented and not all are helpfull. E.g. Table 3 doesn't add a lot for the readership.

In the discussion a number of studies on the same topic performed in Mexico are mentioned. They should be referred to in the introduction already.

Author Response

Las respuestas están en el archivo adjunto.

Reviewer 2 Report

Review regarding "children-2053586-peer-review-v1" manuscript.
This is an interesting cross-sectional study of metabolic syndrome prevalence in Mexican children and adolescents. We always welcome studies of metabolic syndrome prevalence in children and adolescents of various ethnic groups.
Although it is stated in the limitations section, the main problem of this study is the small sample size. Seventy-two subjects is a very small sample size for this kind of study. It represents about 1/10000th of the least estimation of Veracruz state, which is definitely a poor number.
Furthermore, this population came from two public educational institutions. How many public schools are there in Huatusco town? Are there any private schools? Are the 72 children representative of the Mexican population of Veracruz? The authors should answer all the above questions to state this study's selection bias.
The title should be changed to " Prevalence of metabolic syndrome and its relationship with biochemical indicators in the Huatsuo town, Veracruz state, Mexico"
P2L71: From the text, we assume that percentiles were estimated for weight and waist circumference. Which distribution was used? The subject's own distribution or the Mexican ethnic distribution? Later the authors state that BMI Z-score was estimated from the WHO distribution. This small discrepancy should be evident in the text.
P2L88-93: Too much information is written about chemistry kits. Authors should simplify the paragraphs of kits.
P3L115: For the presentation unity it is better to use only non-parametric or only parametric tests after the mathematical transformation of variables.
P3L127: For anthropometric children and adolescent variables it is better to use SDS scores (Z-values) and not actual variables. Otherwise, the differences found, as in the BMI variable, are normal since there are critical but normal differences between boys and girls, which does not need to be stated. The lack of multiple parametric comparisons is evident. Biochemical parameters are depended not only on gender but weight and height. Authors should consider parametric analysis, based on transformed variables in the case of skewness.   
Table 2 and 3 titles should include Huatsuo town because the sample size is extremely small. SDS scores are preferable for the presentation of anthropometric variables.
Table 5 includes cells with fewer than 5 observations (including zero subjects). Fisher's exact test is superior to the chi-square test for such instances.
Table 6. Again it would be best to use SDS scores than actual variables.
Table 6 and discussion. The cardinal finding of this study is the correlation of CRP with metabolic syndrome even from a young age. Subclinical inflammation is the key factor for endothelial dysfunction and this feature should be magnified in the discussion section. Some evidence should be referred here by the authors.
In conclusion section authors should mention the need for national and international obesity prevention programs.  

Author Response

(The authors gave the same response as above.)

Round 2

Reviewer 1 Report

The number of children eligible for the study and the number invited for the study are not mentioned.

Author Response

Las respuestas a sus comentarios están en la nueva versión. Sin embargo, es importante comentar que actualmente estamos trabajando con las observaciones del Editor Académico, donde se han atendido los comentarios al manuscrito.
